# EntProp: High Entropy Propagation for Improving Accuracy and Robustness

Shohei Enomoto[1]

[1]NTT, Tokyo, Japan

## Abstract

Deep neural networks (DNNs) struggle to generalize to out-of-distribution domains that are different from those in training despite their impressive performance. In practical applications, it is important for DNNs to have both high standard accuracy and robustness against out-of-distribution domains. One technique that achieves both of these improvements is disentangled learning with mixture distribution via auxiliary batch normalization layers (ABNs). This technique treats clean and transformed samples as different domains, allowing a DNN to learn better features from mixed domains. However, if we distinguish the domains of the samples based on entropy, we find that some transformed samples are drawn from the same domain as clean samples, and these samples are not completely different domains. To generate samples drawn from a completely different domain than clean samples, we hypothesize that transforming clean high-entropy samples to further increase the entropy generates out-of-distribution samples that are much further away from the in-distribution domain. On the basis of the hypothesis, we propose high entropy propagation (EntProp), which feeds high-entropy samples to the network that uses ABNs. We introduce two techniques, data augmentation and free adversarial training, that increase entropy and bring the sample further away from the in-distribution domain. These techniques do not require additional training costs. Our experimental results show that EntProp achieves higher standard accuracy and robustness with a lower training cost than the baseline methods. In particular, EntProp is highly effective at training on small datasets.

## 1 INTRODUCTION

Deep neural networks (DNNs) have achieved impressive performance in a variety of fields, such as computer vision, natural language processing, and speech recognition. However, DNNs are susceptible to accuracy degradation when presented with data distributions that deviate from the training distribution. This is a common occurrence in outdoor environments, such as autonomous driving and surveillance cameras, due to variations in weather and brightness [Diamond et al., 2021, Hendrycks and Dietterich, 2019, Zendel et al., 2018]. As a result, while standard accuracy is essential for DNNs, robustness against distribution shifts is equally important.

Various techniques have been proposed to improve robustness against out-of-distribution domains (*e.g.*, domain adaptation [Saenko et al., 2010, Ganin and Lempitsky, 2015, Tzeng et al., 2015]), many of which usually decrease the standard accuracy. One technique to improve both standard accuracy and robustness is disentangled learning with mixture distribution using a dual batch normalization (BN) layer [Xie et al., 2020, Mei et al., 2022, Zhang et al., 2022, Wang et al., 2021]. This technique prepares an auxiliary BN layers (ABNs) in addition to the main BN layers (MBNs). It feeds the clean samples and the samples transformed by adversarial attacks or data augmentation to the same network but applied with different BNs, *i.e.*, use the MBNs for the clean samples and use the ABNs for the transformed samples. The distinction of the BNs used to train samples of different domains prevents mixing of the BN layer statistics and the affine parameters [Zhang et al., 2023], allowing the MBN-applied network to learn better from the features of both the out-of-distribution and in-distribution domains [Xie et al., 2020]. Furthermore, since only MBNs are used during inference, there is no increase in computational cost in test-time.

Existing studies treat clean and transformed samples as different domains; however, it is not clear whether these samples are entirely different domains. It is clear that clean

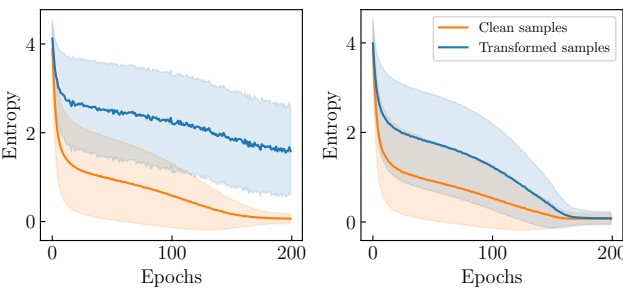

Figure 1: Entropy per epoch when ResNet-18 is trained with MixProp [Zhang et al., 2022] (left) and AdvProp [Xie et al., 2020] (right) on the CIFAR-100 dataset. Error bars indicate one standard deviation, and lines indicate average.

samples are in-distribution domain. The transformed samples can be divided into two groups: those that are highly transformed and those that are less transformed. Therefore, we have the following research questions: *Do transformed samples include samples drawn from both the in-distribution and out-of-distribution domains?*

As a first step in answering this question, we consider distinguishing between the in-distribution and out-of-distribution samples. Since adversarial attacks and data augmentation are transformations that increase the diversity and hardness of samples [Wang et al., 2021], we verify the distinction of domains using an uncertainty metric, entropy. Figure 1 shows the entropy of clean and transformed samples when training the network with baseline methods. The results show that some of the clean samples with high entropy overlap with the entropy of the transformed samples. Since clean high-entropy samples are already similar to out-of-distribution samples, we hypothesize that applying entropy-increasing transformations to clean high-entropy samples generates out-of-distribution samples that are much further away from the in-distribution samples. From this hypothesis, we propose high entropy propagation (EntProp), which trains ABN-applied network with high-entropy samples. First, a network trains clean samples using MBNs and calculates entropy. Then, for the high-entropy samples in the clean samples, a network trains using ABNs. At this time, to further increase the entropy of the samples and bring them further away from the in-distribution domain, we introduce two techniques, data augmentation and free adversarial training [Shafahi et al., 2019]. These techniques have no additional training cost and allow for further accuracy gains.

We evaluated EntProp on five widely used image classification datasets with several DNN architectures. We show that simply training ABN-applied network on clean high-entropy samples improves both standard accuracy and robustness even though it does not use adversarial attacks or data augmentation. EntProp, which includes two entropy-increasing techniques, shows higher accuracy at a lower training cost than baseline methods. Furthermore, we show that on the

small dataset, the use of adversarial training on all samples leads to overfitting, which can be resolved by effective undersampling, such as EntProp.

The contributions of this paper are as follows:

- We propose a novel disentangled learning method via ABNs that distinguishes sample domains based on entropy. We show that training ABN-applied network on high-entropy samples improves both standard accuracy and robustness.

- We introduce two techniques, data augmentation and free adversarial training, which further increase sample entropy and model accuracy without training cost.

- Our extensive experiments show that EntProp achieves better standard accuracy and robustness than baseline methods, despite its lower training cost. We show that on small datasets, using all samples for adversarial training leads to overfitting, while undersampling methods such as EntProp prevent overfitting, benefit from adversarial training, and improve accuracy.

## 2 RELATED WORK

Adversarial attacks [Goodfellow et al., 2015, Madry et al., 2018] cause DNNs to make wrong predictions by adding human imperceptible perturbations to input sample. To defend against such attacks, a variety of methods [Kannan et al., 2018, Zhang et al., 2019, Wang et al., 2020] have been proposed to train DNNs with adversarial samples, also known as adversarial training. However, adversarial training has a trade-off [Tsipras et al., 2018, Ilyas et al., 2019] between accuracy on clean samples and robustness to adversarial attacks, compromising accuracy on clean samples to achieve high robustness. The reason for this trade-off was thought to be that the two domains are learned simultaneously by a single DNN, motivated by the two-domain hypothesis [Xie and Yuille, 2019] that clean and adversarial samples are drawn from different domains. Based on this hypothesis, Xie and Yuille [2019] showed that using MBNs for clean samples and ABNs for adversarial samples avoids mixing the statistics and affine parameters of BN layers [Zhang et al., 2023] by two different domains and achieves high accuracy for the domain for which each BN layer is trained. AdvProp [Xie et al., 2020] showed that disentangled learning for a mixture of distributions via ABNs allows DNNs with MBNs to learn more effectively from both adversarial and clean samples, improving the standard accuracy and the accuracy for the out-of-distribution domain. AdvProp is simple and highly practical, and has since been developed in various ways. Fast AdvProp [Mei et al., 2022] reduced the number of samples and iterations required for adversarial attacks, resulting in the same computational cost as vanilla training, with higher accuracy. Disentangled learning via ABNs showed effectiveness not only using adversarial attacked

samples, but also using data augmented samples [Merchant et al., 2020, Zhang et al., 2022, Wang et al., 2021] and style transferred samples [Li et al., 2020]. Furthermore, AdvProp was proposed for various applications, including object detection tasks [Chen et al., 2021], contrastive learning [Jiang et al., 2020, Ho and Nvasconcelos, 2020], and training vision transformers [Herrmann et al., 2022].

Although these studies treat clean and transformed samples as different domains, we argue that some of these samples overlap in domain. We train the MBN-applied network with in-distribution domain samples and the ABN-applied network with high-entropy samples as the out-of-distribution domain.

# 3 PROPOSED METHOD

In this section, we describe our method, high entropy propagation (EntProp), for effective disentangled learning with mixture distribution via ABNs.

## 3.1 MOTIVATION

Baseline methods treat clean samples as the in-distribution domain and samples transformed by adversarial attacks [Xie et al., 2020, Mei et al., 2022, Xie and Yuille, 2019] or data augmentation [Zhang et al., 2022, Merchant et al., 2020] as the out-of-distribution domain, and distinguish the BNs used for these samples. Although it is clear that clean samples are the in-distribution domain, we question that transformed samples are the out-of-distribution domain. In the transformed samples, some samples are significantly affected by the transformation and are further away from the in-distribution domain, while some samples are less affected and closer to the in-distribution domain. Because the MixUp and PGD attack used by MixProp and AdvProp are both sample transformations that increase entropy, we use entropy as the initial investigation to distinguish the distributions. If the distribution is distinguished by entropy as shown in Figure 1, some samples in the clean and transformed samples have overlapping domain, which may prevent effective disentangled learning via ABNs. Since clean high-entropy samples are in the same domain as the transformed out-of-distribution samples, we hypothesize that transforming these samples to increase entropy generates out-of-distribution samples that are significantly different from the in-distribution samples. On the basis of the hypothesis, we propose EntProp, which trains the ABN-applied network on high-entropy samples.

## 3.2 METHODOLOGY

Here, we describe the process of one iteration of EntProp training. We assume a network with ABNs in addition to the

MBNs. Figure 2 shows the overview of EntProp and baseline methods, and Algorithm 1 shows the pseudo-code of EntProp. EntProp consists of three components, *Sample Selection*, *Data Augmentation* and *Free Adversarial Training*, which we will detail subsequently.

**Sample Selection.** First, the MBN-applied network outputs prediction $\hat{p}(y|x)$ for the class label $y$ from clean sample $x$. From the prediction, we compute the loss and entropy.

$$H = \sum_{y=1}^{C} -\hat{p}(y|x) \log \hat{p}(y|x), \qquad (1)$$

where $H$ is entropy and $C$ is the number of classes. Next, we feed the top $k|\mathcal{B}|$ samples of high-entropy samples to the ABN-applied network to compute the loss, where $k \in [0, 1]$ is a hyperparameter and $|\mathcal{B}|$ is the batch size. Finally, we update the network parameters from the gradient to minimize total loss. Furthermore, based on our hypothesis, we introduce two techniques that increase the entropy of samples without additional training cost: data augmentation and free adversarial training.

**Data Augmentation.** Data augmentation is the most common technique widely used when training DNNs that improves the accuracy of DNNs by transforming samples and increasing diversity and hardness. Since most data augmentations use simple transformations, the computational cost is negligible compared to training DNNs. We use the popular data augmentation technique, MixUp [Zhang et al., 2018], to increase the entropy of the samples. MixUp linearly combines two samples in a mini-batch and increases entropy because the combined sample has two labels. Unlike Mix-Prop [Zhang et al., 2022], we treat augmented samples as in-distribution domain and train MBN-applied network from the augmented samples for the calculation of loss and entropy. Since MixUp improves standard accuracy, samples transformed by MixUp retain sufficient information about the in-distribution domain. Furthermore, MixUp eliminates the high-entropy sample selection bias in each iteration, allowing the ABN-applied network to train a diversity of samples (see Section 4.3.4 for details). The MixUp loss function is defined as:

$$L^m = \lambda L^c(\theta, \boldsymbol{x}^m, y^a) + (1 - \lambda)L^c(\theta, \boldsymbol{x}^m, y^b), \quad (2)$$

where $L^c$ is the cross-entropy loss, $\theta$ is the network parameter, $\lambda$ is the mixing coefficient, $\boldsymbol{x}^m$ is the mixed samples, and $y^a$ and $y^b$ are the labels of the samples before mixing. If EntProp does not use MixUp, the MBN-applied network trains $L^c$ for clean samples.

**Free Adversarial Training.** Shafahi et al. [2019] generates adversarial examples by reusing the gradients used for training in the previous iteration. We use this technique to generate adversarial examples $\boldsymbol{x}^a$ for high-entropy samples. EntProp first calculates the loss to clean or augmented

Table 1: Training costs for each method. $p_{\text{adv}}$ is the Fast AdvProp hyperparameter that determines the sample percentage used for adversarial attack.

| | Vanilla | AdvProp | Fast AdvProp | MixProp | EntProp |
|---|---|---|---|---|---|
| Training Cost | N | $(2+n)$N | $(1+p_{\text{adv}})$N | 2N | $(1+kn)$N |

samples with MBN-applied network, allowing the generation of free adversarial examples from the gradient at this time. Note that it is not optimal to use the MBN-applied network gradient to generate an adversarial attack on the ABN-applied network. When we use augmented samples, we use the gradient obtained from the augmentation loss to generate an adversarial example. In the case of multiple iterations for the attacker, as in a Projected Gradient Descent (PGD) [Madry et al., 2018] attack, the first one has no computational cost, but the subsequent ones have the same computational cost as a standard adversarial attack and are generated from the gradient of the ABN-applied network. For the PGD attack, we set perturbation size $\epsilon$ to $n+1$ and attack step size $\alpha$ to 1, where $n$ is the number of iterations for the attacker. If the number of iterations is 1, then $\epsilon$ is set to 1.

### 3.3 TRAINING COST

Here, we consider the training cost of one epoch. We denote the cost of a single forward and backward pass for a single sample as 1 and the size of the dataset as $N$. The cost of vanilla training for one epoch is $N$. EntProp first uses the clean mini-batch, then $k|\mathcal{B}|$ samples of the mini-batch, thus the cost is $(1+k)N$. The computational cost of data augmentation and free adversarial training ($n=1$) is negligible compared to the computational cost of forward and backward passes, thus using them does not change the overall training cost. If we increase the iteration number $n$ of the adversarial attack by more than 1, it cost us an additional $k(n-1)N$. Consequently, the training cost of EntProp is $(1+kn)N$. Table 1 shows the training cost per epoch for baseline methods and EntProp.

## 4 EXPERIMENTS

In this section, we experimented on five widely used image classification datasets and show the effectiveness of EntProp.

### 4.1 EXPERIMENTS SETUP

#### 4.1.1 Datasets

To extensively evaluate the effectiveness of EntProp and baseline methods, we measure performance across the following five datasets that are widely used for image

---

**Algorithm 1:** Pseudo code of EntProp

**Data:** A set of clean samples with labels;
**Result:** Network parameter $\theta$;
**for** *each training step* **do**
    Sample a clean mini-batch $x$ with label $y$;
    Generate the corresponding augmented mini-batch $x^m$ and labels $y^a$ and $y^b$;
    Compute loss $L^m$ and entropy $H$ on augmented mini-batch using the MBNs from Equations (1) and (2);
    Obtain the gradient $\nabla \leftarrow \nabla_{x^m}$;
    Get the top$k|\mathcal{B}|$ samples $x^a$ with the highest entropy from augmented mini-batch;
    $\delta \leftarrow 0$;
    **for** $i = 1, \ldots, n$ **do**
        $\delta \leftarrow \delta + \epsilon \cdot sign(\nabla)$;
        $x^a = x^a + clip(\delta, -\epsilon, \epsilon)$;
        Compute loss $L^c(\theta, x^a, y)$ on adversarial sample using the ABNs;
        Obtain the gradient $\nabla \leftarrow \nabla_{x^a}$;
    **end**
    Minimize the total loss w.r.t. network parameter $\underset{\theta}{\arg\min}\, L^m + L^c(\theta, x^a, y)$.
**end**
**return** $\theta$

---

classification benchmark: CIFAR-100 (C100) [Krizhevsky et al., 2009], CUB-200-2011 (CUB) [Welinder et al., 2010], OxfordPets (Pets) [Parkhi et al., 2012], Stanford-Cars (Cars) [Krause et al., 2013] and ImageNet (IN) [Russakovsky et al., 2015]. We provide details on each dataset in the Appendix.

#### 4.1.2 Comparison Methods.

We compared the four baseline methods with EntProp.

- **Vanilla.** Vanilla training for network without ABNs.
- **AdvProp.** AdvProp feeds the clean samples and the adversarial samples to the same network but applied with different BNs. We used PGD as the attacker to generate adversarial samples. We set the perturbation size $\epsilon$ to 4. The number of iterations for the attacker is $n=5$ and the attack step size is $\alpha=1$.
- **Fast AdvProp.** Fast AdvProp speeds up AdvProp by reducing the number of iterations for PGD attacker and the percentage of training samples used as adversarial examples. We set the percentages of training samples used as adversarial examples to $p_{adv}=0.2$, the perturbation size $\epsilon$ to 1, the number of iterations for the attacker to $n=1$, and the attack step size to $\alpha=1$.
- **MixProp.** MixProp feed the clean samples and the augmented samples with MixUp to the same network but

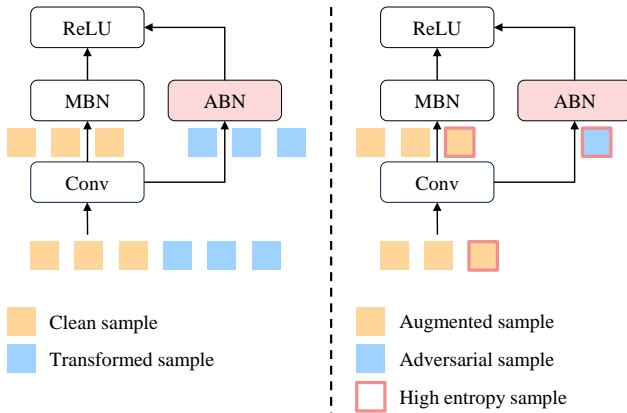

Figure 2: Overview of baseline methods (left) and EntProp (right). The baseline methods feed clean samples to MBN and transformed samples to ABN. EntProp treats the augmented sample as in-distribution domain and feeds it to MBN. EntProp then adversarial attacks high-entropy samples and feeds it to ABN.

applied with different BNs. The parameter of the beta distribution used for MixUp is set to 0.2 for ImageNet and 1 otherwise.

### 4.1.3 Implementation Details

For the C100 experiments, we trained DNNs for 200 epochs with a batch size of 128, using SGD with momentum of 0.9 and weight decay of 0.0005. The learning rate started with 0.1 and decreased by cosine scheduler. For the CUB, Pets, and Cars experiments, we fine-tuned the DNNs, which were pre-trained [maintainers and contributors, 2016] on the ImageNet dataset, using Adam [Kingma and Ba, 2015] optimizer. Since the pretrained DNNs do not have ABNs, we set the initial weights of the ABNs to be the same as those of the MBNs. We fine-tuned networks with batch size of 64 for 100 epochs with weight decay of 0.0005. The learning rate started with 0.0001 and decreased by the factor of 0.1 at every 10 epochs. For the IN experiments, we trained DNNs for 105 epochs with a batch size of 256, using SGD with momentum of 0.9 and weight decay of 0.0005. The learning rate started with 0.1 and decreased by the factor of 0.1 at every 30 epochs.

In accordance with the the Fast AdvProp setting, we adjust the baseline methods to align the loss scale with that of vanilla training. Furthermore, given that Fast AdvProp and EntProp entail the duplication of training instances within a single iteration, we ensure equitable treatment of all samples through weight normalization.

### 4.1.4 Evaluation Metrics.

We evaluate standard accuracy (SA), the accuracy of a standard test set, and robust accuracy (RA), the average accuracy of an artificially corrupted test set [Hendrycks and Dietterich, 2019]. Artificial corruptions are the same as those used in ImageNet-C dataset and the corrupted test set consists of 15 types of corruption with five severity levels, and we use the average accuracy of all of them as RA. Furthermore, to evaluate the balance between SA and RA, we define the harmonic mean as our evaluation metric.

$$H_{score} = \frac{2SA \cdot RA}{SA + RA}. \qquad (3)$$

$H_{score}$ is high only when both SA and RA are high.

## 4.2 MAIN EXPERIMENTS

In this section, we show the effectiveness of EntProp. We describe more detailed experimental results in the Appendix. All experiments, except the ImageNet experiment, were performed three times, and we report the average values. The best and second results are **bolded** and underlined.

### 4.2.1 Comparison Results

Table 2 shows the comparison results between EntProp and the baseline methods. EntProp ($k = 0.2, n = 1$) and EntProp ($k = 0.2, n = 5$) have the same training cost as Fast AdvProp and MixProp, respectively, but outperform $H_{score}$. EntProp ($k = 0.6, n = 5$) has a much lower training cost than AdvProp, but shows the highest $H_{score}$ for all datasets except Cars. These results indicate that EntProp allows for more efficient training by bringing the samples fed to the ABN-applied network further away from the in-distribution domain. For small datasets such as CUB and Pets, EntProp shows particularly large improvement results, while AdvProp shows smaller improvement results. Because adversarial training requires large datasets [Schmidt et al., 2018], AdvProp leads to overfitting [Rice et al., 2020] on small datasets, making it difficult to improve performance. EntProp mitigates overfitting issues by employing efficient entropy-based undersampling techniques, thereby achieving notable accuracy improvements through effective adversarial training strategies.

Moreover, in Figure 3 we demonstrate the trade-off between average $H_{score}$ and training cost across all datasets excluding ImageNet. EntProp shows a higher $H_{score}$ with a smaller increase in training cost and a better trade-off than the baseline methods.

Table 2: Accuracy and training cost of training ResNet-50 with each method on five datasets. † indicates that it is a number from the original literature.

| Method | Dataset Cost | C100 SA(%) | RA(%) | H$_{score}$ | CUB SA(%) | RA(%) | H$_{score}$ | Pets SA(%) | RA(%) | H$_{score}$ | Cars SA(%) | RA(%) | H$_{score}$ | IN SA(%) | RA(%) | H$_{score}$ |
|---|---|---|---|---|---|---|---|---|---|---|---|---|---|---|---|---|
| Vanilla | N | 79.30 | 51.01 | 62.08 | 81.99 | 48.37 | 60.85 | 92.24 | 49.97 | 64.82 | 90.18 | 41.07 | 56.43 | 76.13 | 39.57 | 52.07 |
| MixProp | 2N | **81.84** | 55.55 | 66.18 | **83.77** | 56.80 | 67.70 | **93.01** | 59.84 | 72.82 | **91.30** | 51.13 | 65.55 | **77.20** | 41.79 | 54.23 |
| Fast AdvProp | 1.2N | 79.43 | 53.31 | 64.45 | 82.90 | 51.22 | 63.32 | 92.78 | 54.42 | 68.60 | 90.71 | 44.31 | 59.54 | 76.60 | 40.71 | 53.16 |
| AdvProp | 7N | 78.05 | 58.94 | 67.17 | 81.45 | 53.44 | 64.54 | 92.10 | 55.82 | 69.51 | 90.45 | 54.11 | 67.71 | 77.10† | N/A | N/A |
| EntProp ($k = 0.2, n = 1$) | 1.2N | 79.99 | 56.07 | 65.93 | 82.92 | 60.18 | 69.74 | 92.23 | 59.28 | 72.18 | 90.48 | 52.93 | 66.79 | 76.29 | 42.70 | 54.75 |
| EntProp ($k = 0.6, n = 1$) | 1.6N | 80.31 | 57.41 | 66.12 | 83.32 | 62.02 | 71.11 | 92.47 | 62.35 | 74.48 | 90.15 | **55.66** | 68.82 | 75.09 | 41.49 | 53.45 |
| EntProp ($k = 0.2, n = 5$) | 2N | 78.21 | 57.27 | 66.95 | 83.10 | 60.71 | 70.17 | 92.21 | 60.14 | 72.80 | 90.36 | 52.00 | 66.01 | 76.35 | 43.13 | 55.12 |
| EntProp ($k = 0.6, n = 5$) | 4N | 80.62 | **60.50** | **69.12** | 82.65 | **64.00** | **72.14** | 92.15 | **66.75** | **77.42** | 90.21 | 52.72 | 66.55 | 76.47 | **44.45** | **56.22** |

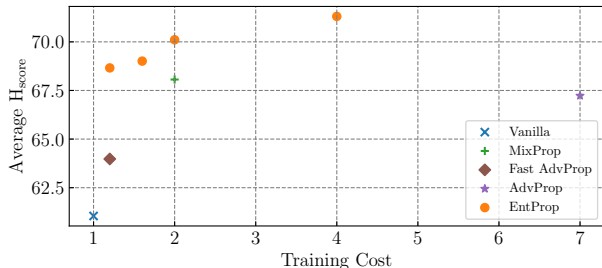

Figure 3: Average H$_{score}$ and training cost over all datasets except ImageNet. We plot the relative values with the vanilla training cost as 1.

Table 3: Accuracy on distribution-shifted datasets other than the corrupted dataset when ResNet-50 is trained with each method. A, R, and Stylized denote ImageNet-A [Hendrycks et al., 2021b], ImageNet-R [Hendrycks et al., 2021a], and Stylized-ImageNet [Geirhos et al., 2018], respectively.

| Method | A | R | Stylized |
|---|---|---|---|
| Vanilla | 0.00 | 36.17 | 7.18 |
| MixProp | 3.17 | 38.75 | 8.32 |
| Fast AdvProp | 2.19 | 38.17 | 8.17 |
| EntProp ($k = 0.2, n = 1$) | 2.87 | 39.88 | 9.56 |
| EntProp ($k = 0.6, n = 1$) | 2.60 | 38.85 | 9.73 |
| EntProp ($k = 0.2, n = 5$) | 2.89 | 39.85 | 10.69 |
| EntProp ($k = 0.6, n = 5$) | **3.29** | **40.78** | **10.94** |

#### 4.2.2 Other Distribution Shift Datasets

Here, we evaluate EntProp on distribution-shifted datasets other than the corrupted dataset. Table 3 shows the accuracy of EntProp and baseline methods on the ImageNet variant datasets. Disentangled learning methods using ABN improve accuracy even under various types of distribution shifts, with EntProp showing the highest accuracy among them.

#### 4.2.3 Other Architectures

Table 4 shows experimental results for several architectures on the CIFAR-100 dataset. Regardless of architecture, EntProp ($k = 0.6, n = 5$) consistently shows the highest H$_{score}$.

Furthermore, we investigated with the applicability of EntProp to vision transformers (ViT). We experimented with fine-tuning ViT-Base pre-trained by MAE [He et al., 2022] on the CIFAR-100 dataset. When applying EntProp to ViT, we add an auxiliary layer normalization layer instead of an auxiliary BN layer. We show the results in Table 5. The results show that EntProp improves the SA and RA of ViT. EntProp can be applied to ViT-based architectures and can be used in conjunction with recent methods to improve SA and RA such as MAE.

#### 4.2.4 Verification of hypothesis

To verify our hypothesis, we measured Frechet Inception Distance (FID), a measure of inter-distributional distance between two datasets. We used Vanilla-trained ResNet-18 on the CIFAR-100 dataset. We provide FID for the original dataset and the dataset generated by three transformations: MixUp , PGD attack, and Ours (Sample selection + PGD attack + MixUp). Table 6 shows the results. Ours shows the most FID increase and results in the pushing original distribution far away from another. This result supports our hypothesis that transforming samples to increase entropy generates out-of-distribution samples.

#### 4.2.5 Ablation Study

First, we do not use data augmentation and free adversarial training, and we confirm the effect of feeding only clean high-entropy samples to the ABN-applied network. Figure 4 shows the results of sample selection with high entropy versus random selection. There is little difference when $k$ is small and large, and entropy shows higher H$_{score}$ than random when $k = 0.2$ to $k = 0.7$. Furthermore, $k \geq 0.1$ shows a higher H$_{score}$ than vanilla training ($k = 0$), meaning that the use of ABN is effective. The use of ABN increases the

Table 4: Accuracy of ResNet-18, WideResNet-50, and ResNeXt-50 trained on the CIFAR-100 dataset. Avg. indicates the average of the three networks.

| Method | ResNet-18 | | | WRN-50 | | | ResNeXt-50 | | | Avg. | | |
|---|---|---|---|---|---|---|---|---|---|---|---|---|
| | SA(%) | RA(%) | $H_{score}$ | SA(%) | RA(%) | $H_{score}$ | SA(%) | RA(%) | $H_{score}$ | SA(%) | RA(%) | $H_{score}$ |
| Vanilla | 78.45 | 49.96 | 61.04 | 79.35 | 51.64 | 62.56 | 80.86 | 52.95 | 63.99 | 79.55 | 51.52 | 62.53 |
| MixProp | **80.86** | 53.97 | 64.73 | **82.17** | 56.38 | 66.87 | **82.37** | 56.97 | 67.36 | **81.80** | 55.77 | 66.32 |
| Fast AdvProp | 78.89 | 53.31 | 63.63 | 79.69 | 55.25 | 65.25 | 79.30 | 55.31 | 65.17 | 79.29 | 54.62 | 64.68 |
| AdvProp | 75.15 | 56.78 | 64.69 | 77.50 | 59.28 | 67.17 | 78.36 | 59.08 | 67.37 | 77.00 | 58.38 | 66.41 |
| EntProp ($k=0.2, n=1$) | 79.41 | 55.24 | 65.15 | 80.66 | 57.30 | 67.00 | 81.46 | 58.47 | 68.08 | 80.51 | 57.00 | 66.74 |
| EntProp ($k=0.6, n=1$) | 78.89 | 55.86 | 65.40 | 81.3 | 58.95 | 68.34 | 81.75 | 59.28 | 68.72 | 80.65 | 58.03 | 67.49 |
| EntProp ($k=0.2, n=5$) | 79.19 | 54.52 | 64.58 | 81.13 | 58.00 | 67.64 | 81.20 | 58.95 | 68.31 | 80.51 | 57.16 | 66.84 |
| EntProp ($k=0.6, n=5$) | 78.92 | **57.16** | **66.30** | 80.77 | **61.02** | **69.52** | 81.38 | **61.35** | **69.96** | 80.36 | **59.84** | **68.59** |

Table 5: Accuracy of ViT-base pre-trained by MAE and fine-tuned on the CIFAR-100 dataset.

| Method | SA(%) | RA(%) | $H_{score}$ |
|---|---|---|---|
| Vanilla | 89.55 | 70.62 | 78.97 |
| EntProp ($k=0.5, n=5$) | **89.56** | **75.39** | **81.87** |

Table 6: FID for the original and transformed datasets. We measured FID using Vanilla-trained ResNet-18 on the CIFAR-100 dataset.

| Transformation | FID |
|---|---|
| MixUp | 5.12 |
| PGD attack | 373.93 |
| Ours | **383.89** |

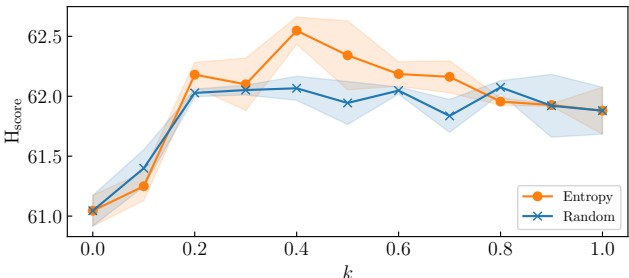

Figure 4: Comparison of high-entropy sample selection to random selection using ResNet-18 on the CIFAR-100 dataset. Error bars indicate one standard error, and lines indicate the average. $k = 0$ is the same as vanilla training, and $k = 1$ feeds all samples to the ABN-applied network.

Table 7: Ablation study with ResNet-18 on the CIFAR-100 dataset. The numbers in parentheses indicate the differences from vanilla training.

| Sample Selection | MixUp | Free ($n=1$) | SA(%) | RA(%) | $H_{score}$ |
|---|---|---|---|---|---|
| ✓ | | | 79.24(0.79) | 51.17(1.21) | 62.18(1.14) |
| ✓ | ✓ | | **79.66**(1.21) | 54.53(4.57) | 64.74(3.70) |
| ✓ | | ✓ | 78.55(0.10) | 52.99(3.03) | 63.29(2.25) |
| ✓ | ✓ | ✓ | 79.41(0.96) | **55.24**(5.28) | **65.15**(4.11) |

number of network parameters during training and allows the network to achieve good generalization performance.

Next, we verified each component of EntProp to confirm the effect of increasing entropy. We set $k = 0.2$ and $n = 1$. Table 7 shows the results. Training clean high-entropy samples with the ABN-applied network improves both SA and RA from vanilla training even though no additional processing, such as adversarial attacks, is performed. MixUp further improves both SA and RA, while free adversarial training further improves RA but slightly decreases SA. EntProp which uses all components achieves the highest $H_{score}$. Increasing entropy brings the sample further away from the in-distribution domain, allowing effective disentangled learning with mixture distribution. Moreover, Figure 5 shows the entropy of the clean and transformed samples when training the network with EntProp. The results show that EntProp ($k = 0.2, n = 5$) completely distinguishes between the domains of clean and transformed samples, as we hypothesize.

## 4.3 DETAILED EXPERIMENTS

In this section, we provide a detailed analysis of the validity of EntProp's design.

### 4.3.1 Uncertainty Metric

We use entropy as a metric to select the samples that EntProp feeds to the ABN-applied network. We evaluated EntProp ($k = 0.2, n = 1$) when using the following uncertainty metrics, in addition to entropy, to distinguish between samples in the in-distribution and out-of-distribution domains.

- **Cross-Entropy** is the distance between the true probability distribution and the predicted probability distribution.
- **Confidence** is the maximum class probability.

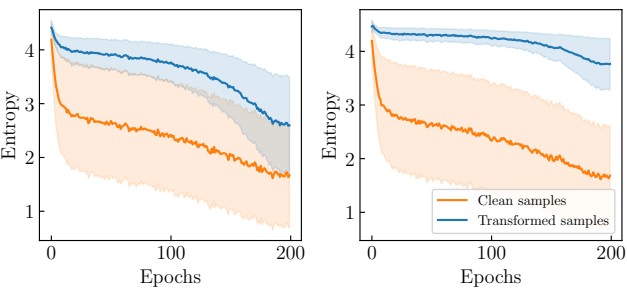

Figure 5: Entropy per epoch when ResNet-18 is trained with EntProp ($k = 0.2, n = 1$) (left) and EntProp ($k = 0.2, n = 5$) (right) on the CIFAR-100 dataset. Error bars indicate one standard deviation, and lines indicate average.

Table 8: $H_{score}$ for different uncertainty metrics on the CIFAR-100 dataset.

| Metrics | ResNet-18 | ResNet-50 | WRN-50 | ResNeXt-50 | Avg. |
|---|---|---|---|---|---|
| Entropy | 65.15 | 65.93 | 67.00 | 68.08 | 66.54 |
| Cross-Entropy | 64.81 | 64.76 | 67.12 | 68.34 | 66.26 |
| Confidence | **65.48** | **66.47** | **67.36** | 67.23 | **66.63** |
| Logit Margin | 64.84 | 66.18 | 65.71 | **68.53** | 66.31 |

- **Logit Margin** is the difference between the maximum non-true class probability and the true class probability.

Because we use MixUp during training, the true label used by these metrics is the original true label of the sample. Table 8 shows the results. All metrics show no significant differences. The results show that different architectures have different effective metrics.

### 4.3.2 Design of Data Augmentation

We compared MixUp and CutMix [Yun et al., 2019] as data augmentations that increase entropy at no additional training cost. CutMix replaces a part of an image with another image, so it has two labels, similar to MixUp. Table 9 shows the results. The results show that MixUp significantly outperforms CutMix in RA and $H_{score}$. MixUp, which transforms the entire image, is more likely to increase entropy than CutMix, which transforms a portion of the image and contributes to improving $H_{score}$. On the other hand, Cut-Mix shows higher SA than MixUp and baseline methods (see Table 4). Therefore, we use MixUp when the goal is to improve $H_{score}$ and CutMix when the goal is to improve SA.

### 4.3.3 Adversarial Attacks Other than PGD

We investigated the influence of adversarial attacks other than PGD. Xie et al. [2020] experimented with the PGD variants GD and I-FGSM and concluded that the type of attack has no effect on performance. We experimented with C&W [Carlini and Wagner, 2017] and TRADES [Zhang

Table 9: $H_{score}$ for different data augmentations with ResNet-18 trained by EntProp ($k = 0.2, n = 1$) on the CIFAR-100 dataset.

| Data Augmentation | SA(%) | RA(%) | $H_{score}$ |
|---|---|---|---|
| MixUp | 79.41 | **55.24** | **65.15** |
| CutMix | **81.39** | 50.78 | 62.54 |

Table 10: Accuracy of ResNet-50 on the CIFAR-100 dataset when changing adversarial attacks used by EntProp.

| Adversarial Attack | SA(%) | RA(%) | $H_{score}$ |
|---|---|---|---|
| PGD | **80.62** | **60.50** | **69.12** |
| C&W | 79.93 | 55.86 | 65.76 |
| TRADES | 80.07 | 58.25 | 67.44 |
| TRADES w/Free | 79.81 | 58.13 | 67.27 |

et al., 2019], which are different types of attacks from PGD. We trained ResNet-50 on the CIFAR-100 dataset with Ent-Prop ($k = 0.6, n = 5$) using different attacks. We show the results in Table 10. C&W and TRADES results are inferior to PGD for both SA and RA. C&W is an attack designed to increase DNN misclassification and TRADES is an attack designed to balance DNN accuracy and adversarial robustness, and has a smaller effect on increasing entropy than PGD. Thus, this result confirms the validity of our method design, which uses transformations that increase entropy. In addition, we verify the effectiveness of free adversarial training against other kinds of attack. We experimented with introducing free adversarial training into TRADES, an adversarial attack that uses KL-distance loss in addition to cross-entropy loss. As a result, free adversarial training slightly reduces accuracy, but also reduces training cost by N. For TRADES, free adversarial training using only cross-entropy is not optimal, but it is effective enough in our research context. Thus, free adversarial training generalizes across different kinds of adversarial attacks.

### 4.3.4 Sample Selection Bias

We verified the bias of high-entropy sample selection during training. Figure 6 shows the results. At $k = 0.2$, the bias is large and most samples are not selected as high-entropy samples. MixUp eliminates high-entropy sample selection bias. The bias decreases as $k$ increases, but MixUp shows the effect of further reducing the bias.

### 4.3.5 How to Determine Hyperparameter

EntProp has two hyperparameters: $k$, which determines the percentage of sample fed to the ABN-applied network, and $n$, which is the number of iterations of the PGD attack. These values are determined based on the computational

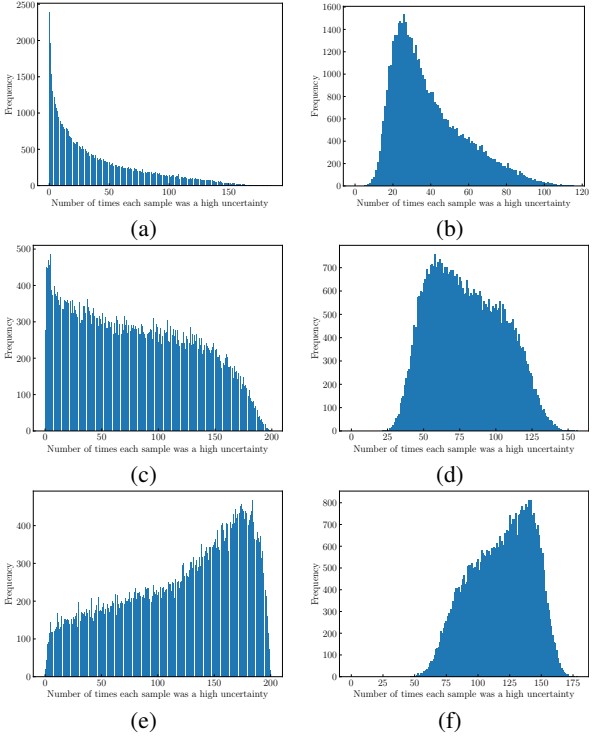

Figure 6: Histogram of the number of times a sample was selected as a high-entropy sample. The vertical axis is frequency and the horizontal axis is number of times each sample was a high uncertainty. We trained ResNet-18 on the CIFAR-100 dataset with different values of k and with and without MixUp. (a) $k = 0.2$. (b) $k = 0.2$ w/MixUp. (c) $k = 0.4$. (d) $k = 0.4$ w/MixUp. (e) $k = 0.6$. (f) $k = 0.6$ w/MixUp.

budget or on the validation accuracy. As shown in the results of the main experiments, large $k$ and $n$ are not the best. We provide the accuracies for varying $k$ and $n$ in the Appendix.

## 5 LIMITATION

In this paper, we focus on improving both standard accuracy and robustness against out-of-distribution domains. We additionally evaluated the robustness against the adversarial attack. We evaluated the accuracy of EntProp variants and vanilla training against PGD-20 attack. Table 11 shows the results. Feeding clean high-entropy samples to the ABN-applied network shows higher adversarial robustness than vanilla training, even though adversarial attacks are not used for training. Free adversarial training significantly improves adversarial robustness, but MixUp significantly decreases it. Comparison of sample selection metrics shows little difference in results across uncertainty metrics. These results indicate that each component of EntProp designed on entropy is effective in improving standard accuracy and out-of-distribution robustness; however, it is not effective in

Table 11: Adversarial robustness of ResNet-18 on the CIFAR-100 dataset.

| Sample Selection | MixUp | Free ($n = 1$) | Metric | PGD-20 |
|---|---|---|---|---|
| | | | | 6.14 |
| ✓ | | | Entropy | 6.44 |
| ✓ | ✓ | | Entropy | 4.14 |
| ✓ | | ✓ | Entropy | 10.51 |
| ✓ | ✓ | ✓ | Entropy | 4.71 |
| ✓ | ✓ | ✓ | Cross-Entropy | 4.45 |
| ✓ | ✓ | ✓ | Confidence | 4.59 |
| ✓ | ✓ | ✓ | Logit Margin | 4.42 |

improving adversarial robustness. If the objective is a different evaluation metric than ours, it is necessary to design an appropriate metric that is different from the entropy.

## 6 CONCLUSION

The existing disentangled learning methods train from mixture distribution by treating clean and transformed samples as different domains, and feeding the former to the MBN-applied network and the latter to the ABN-applied network. However, it is not appropriate to treat the clean and transformed samples as different domains. We found that when we verified the domains of the samples based on entropy, the clean and transformed samples had overlapping regions of domains. We hypothesize that further increasing the entropy of clean high-entropy samples generates samples that are further away from the in-distribution domain. On the basis of the hypothesis, we propose a novel method, EntProp, which feeds high-entropy samples to the ABN-applied network. Our experiments show that EntProp has high accuracy, although its training cost is less than that of baseline methods. In particular, experiments on the small datasets show that Entprop prevents overfitting against adversarial training and outperforms comparison methods. Our method improves standard accuracy and out-of-distribution robustness, but has limitations with respect to adversarial robustness. This limitation suggests the need to design an optimal domain selection metric for each task.

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

# EntProp: High Entropy Propagation for Improving Accuracy and Robustness (Supplementary Material)

**Shohei Enomoto**[1]

[1]NTT, Tokyo, Japan

The supplementary materials for "EntProp: High Entropy Propagation for Improving Accuracy and Robustness".

## A  DATASET DETAILS

**CIFAR-100.**   CIFAR-100 dataset consists of 50000 training images and 10000 test images, with 100 classes.

**CUB-200-2011.**   CUB-200-2011 dataset consists of 5994 training images and 5794 test images, with 200 classes.

**OxfordPets.**   OxfordPets dataset consists of 3669 training images and 3680 test images, with 37 classes.

**StanfordCars.**   StanfordCars dataset consists of 8144 training images and 8041 test images, with 196 classes.

**ImageNet.**   ImageNet dataset consists of 1.3 million training images and 50000 test images, with 1000 classes.

## B  COMPARISON WITH GPACO

GPaCo [Cui et al., 2023] is a loss function that improves both SA and RA. We combined EntProp with GPaCo and experimented using ViT-base pre-trained by MAE on the CIFAR-100 dataset. Table 12 shows the results. The results show that $H_{score}$ improves when used in conjunction with EntProp. EntProp can be used in conjunction with recent methods to improve $H_{score}$.

## C  IMPROVEMENTS THROUGH MIXUP

The performance gain from MixUp is significant, but EntProp's performance does not only come from MixUp. We used Fast AdvProp and MixUp to train the model on the CIFAR-100 dataset and compare it to EntProp. We show the results in Table 13. In all models, EntProp outperforms Fast AdvProp in SA and RA. The most significant difference between EntProp and Fast AdvProp is sample selection. EntProp is entropy-based, while Fast AdvProp randomly selects samples.

Table 12: Accuracy of GPaCo and EntProp combinations. We fine-tuned ViT-base pre-trained by MAE on the CIFAR-100 dataset.

| Method | SA(%) | RA(%) | $H_{score}$ |
|---|---|---|---|
| GPaCo | **89.73** | 70.23 | 78.79 |
| GPaCo w/EntProp($k = 0.5, n = 5$) | 89.37 | **70.92** | **79.08** |

Table 13: Comparison of performance gains from MixUp on the CIFAR-100 dataset.

| Method | ResNet-18 | | | WRN-50 | | | ResNeXt-50 | | |
|---|---|---|---|---|---|---|---|---|---|
| | SA(%) | RA(%) | $H_{score}$ | SA(%) | RA(%) | $H_{score}$ | SA(%) | RA(%) | $H_{score}$ |
| Fast AdvProp (w/MixUp) | 78.94 | 54.65 | 64.59 | 80.19 | 57.16 | 66.74 | 80.74 | 57.83 | 67.39 |
| EntProp ($k = 0.2, n = 1$) | **79.41** | **55.24** | **65.15** | **80.66** | **57.30** | **67.00** | **81.46** | **58.47** | **68.08** |

Table 14: Accuracy of Fast AdvProp and EntProp combinations. We trained ResNet-18 on the CIFAR-100 dataset.

| Method | Cost | SA(%) | RA(%) | $H_{score}$ |
|---|---|---|---|---|
| Vanilla | N | 78.45 | 49.96 | 61.04 |
| Fast AdvProp | 1.2N | 78.89 | 53.31 | 63.63 |
| Fast AdvProp w/EntProp ($k = 0.2$) | 1.4N | **79.39** | **55.20** | **65.12** |

Entropy-based sample selection works well together because it further accelerates the entropy increase due to MixUp. Thus, the improvement we claimed comes from MixUp and entropy-based sample selection.

## D   COMBINATION OF FAST ADVPROP/ADVPROP AND ENTPROP

EntProp (w/o Free Adversarial Training) can be combined with Fast AdvProp/Advprop to improve accuracy. We experimented with the combination of Fast AdvProp and EntProp ($k = 0.2$) with ResNet-18 on the CIFAR-100 dataset. When combined with EntProp, Fast AdvProp uses entropy-based sample selection instead of random sample selection. We show the results in Table 14. Combination with EntProp slightly increases computational cost due to entropy calculation overhead, but also increases accuracy. However, it is inferior to EntProp (1.2N) (see Table 4), thus our method design is superior.

Next, we experimented with the combination of AdvProp and EntProp ($k = 0.6$) with ResNet-50 on the CIFAR-100 dataset. When combined with EntProp, AdvProp uses pure PGD attack without Free Adversarial Training. We show the results in Table 15. EntProp reduces the cost of AdvProp and improves accuracy.

## E   ENTROPY PER EPOCH OF ENTPROP VARIANTS

Figure 7 shows the entropy of the clean and transformed samples when training the network with EntProp variant. Two techniques show that they increase the entropy of the sample.

## F   HYPERPARAMETER SENSITIVITY

We evaluated the relationship between the hyperparameters of EntProp, $k$ and $n$, and accuracy. Tables 16 and 17 show the results. EntProp with $k = 0.6$ shows the best $H_{score}$ than $k = 1.0$, which feeds all samples to ABNs. However, a larger $k$ shows higher adversarial robustness, with $k = 1.0$ showing the best results. Feeding all samples to ABNs leads to overfitting for adversarial attacks. To improve robustness against out-of-distribution domains, it is effective to feed ABNs with carefully selected samples. EntProp shows the highest result when the number of iterations $n$ of PDG attacks is 4. The optimal $n$ depends on the size of the network and dataset.

Table 15: Accuracy of AdvProp and EntProp combinations. We trained ResNet-50 on the CIFAR-100 dataset.

| Method | Cost | SA(%) | RA(%) | $H_{score}$ |
|---|---|---|---|---|
| Vanilla | N | 79.30 | 51.01 | 62.08 |
| AdvProp | 7N | 78.05 | 58.94 | 67.17 |
| AdvProp w/EntProp ($k = 0.6$) | 4.6N | **80.85** | **60.68** | **69.33** |

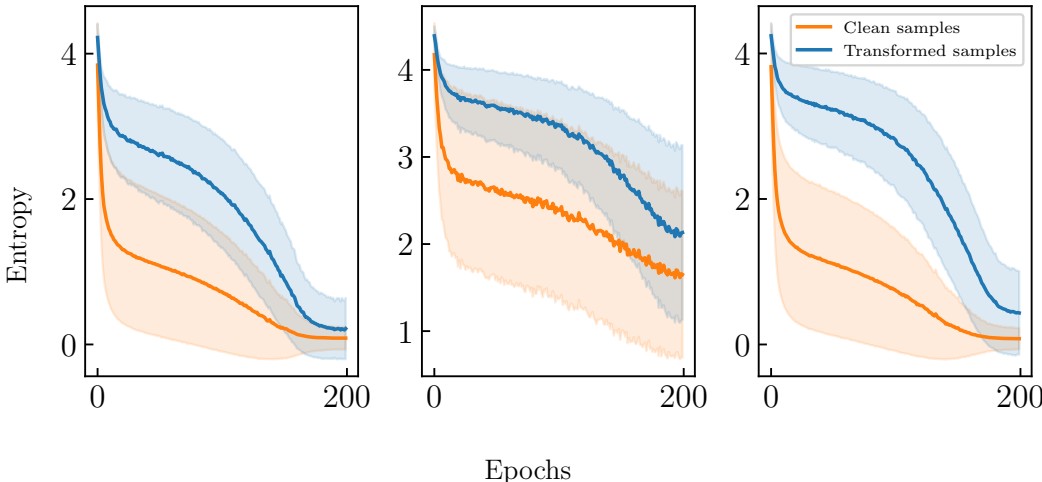

Figure 7: Entropy per epoch when ResNet-18 is trained with EntProp (w/o MixUp, w/o Free adversarial training) (left), EntProp (w/o Free adversarial training) (center), and EntProp (w/o MixUp) (right) on the CIFAR-100 dataset. Error bars indicate one standard deviation, and lines indicate average.

Table 16: Hyperparameter $k$ sensitivity study using ResNet18 on the CIFAR-100 dataset.

| $k$ | SA(%) | RA(%) | $H_{score}$ | PGD-20 |
|-----|-------|-------|-------------|--------|
| 0   | 78.45 | 49.96 | 61.04 | 6.14 |
| 0.1 | 79.15 | 54.30 | 64.41 | 3.89 |
| 0.2 | 79.41 | 55.24 | 65.15 | 4.71 |
| 0.3 | **79.55** | 55.42 | 65.32 | 5.14 |
| 0.4 | 78.90 | 55.55 | 65.20 | 5.14 |
| 0.5 | 79.28 | 56.41 | 65.92 | 6.17 |
| 0.6 | 79.52 | **56.66** | **66.17** | 6.09 |
| 0.7 | 79.12 | 56.04 | 65.61 | 5.96 |
| 0.8 | 79.24 | 56.36 | 65.87 | 6.58 |
| 0.9 | 78.95 | 56.20 | 65.66 | 6.82 |
| 1.0 | 79.44 | 56.61 | 66.11 | **7.25** |

Table 17: Hyperparameter $n$ sensitivity study using ResNet18 on the CIFAR-100 dataset.

| n | SA(%) | RA(%) | $H_{score}$ |
|---|-------|-------|-------------|
| 1 | 78.89 | 55.86 | 65.40 |
| 2 | **79.54** | 57.46 | 66.72 |
| 3 | 79.15 | 57.25 | 66.44 |
| 4 | 79.40 | **58.04** | **67.06** |
| 5 | 78.92 | 57.16 | 66.30 |