# OpenReview forum: "EntProp: High Entropy Propagation for Improving Accuracy and Robustness"
_auai.org/UAI/2024/Conference — UAI 2024 poster_

### Official Review · Reviewer_5ua7 · 2024-03-04

**Q2-1 Originality-Novelty:** 2
**Q2-2 Correctness-Technical Quality:** 2
**Q2-5 Clarity Of Writing:** 3

**Q1 Summary And Contributions:**

This paper proposes a novel entropy-based disentangled learning approach to prevent overfitting, termed EntProp, which identifies high-entropy samples in clean sets and feeds identified samples to ABNs. This paper furthermore proposes two low-cost techniques, data augmentation and free adversarial training, to increase sample entropy. The key idea behind EntProp is that through increasing entropy, high-entropy clean samples can be transformed to out-of-distribution samples, which are much further away from the in-distribution samples.

**Q2-3 Extent To Which Claims Are Supported By Evidence:**

3: Good: the main claims are supported by convincing evidence (in the form of adequate experimental evaluation, proofs, (pseudo-)code, references, assumptions).

**Q2-4 Reproducibility:**

3: Good: key resources (e.g. proofs, code, data) are available and key details (e.g. proofs, experimental setup) are sufficiently well-described for competent researchers to confidently reproduce the main results.

**Q3 Main Strengths:**

This paper innovatively points out that clean samples and transformed samples have overlap in the respect of entropy, which may lead to robust performance degeneration.
This paper includes a fairly comprehensive evaluation of the proposed framework which compares with representative approaches and corresponding insights from results.

**Q4 Main Weakness:**

Entropy is typically used to measure uncertainty in distributions, yet authors propose to increase entropy to reduce the similarities between clean high-entropy samples and out-of-distribution samples, which goes beyond the conventional role of entropy as a measurement. Maximizing entropy might not result in pushing one distribution far away from another, and a clear relationship between the two is lacking. So I suggest that the authors should explain the reason why entropy is chosen for sample distinguishing, or provide further results in the distribution shift caused by the proposed method.
If data augmentation results vary, as indicated in Fig. 5, the main results in Tab. 2 and Tab. 4 should include statistical tests.

**Q5 Detailed Comments To The Authors:**

Fig. 2 is a bit confusing, since the left and the right both have orange and blue boxes, yet the legends are different, which is not intuitive for comparison.
The caption of Fig. 2 has typos.
The x-axis in Fig. 6 is confusing.

**Q9 Complying With Reviewing Instructions:**

Yes

---

> ### Author Rebuttal · Authors · 2024-04-07
>
> We would like to thank you for your in-depth understanding of our paper and your thoughtful feedback to improve its quality.
> We appreciate your recognition that our paper makes an innovative point about the entropy overlap between clean samples and transformed samples, and that our paper includes a comprehensive evaluation.
>
> > Entropy is typically used to measure uncertainty in distributions, yet authors propose to increase entropy to reduce the similarities between clean high-entropy samples and out-of-distribution samples, which goes beyond the conventional role of entropy as a measurement. Maximizing entropy might not result in pushing one distribution far away from another, and a clear relationship between the two is lacking.
>
> As pointed out, maximization entropy might not result in pushing one distribution far away from another.
> Although not theoretical, we have provided empirical insights from extensive experiments, showing that entropy maximization does indeed improve accuracy.
>
> > So I suggest that the authors should explain the reason why entropy is chosen for sample distinguishing
>
> We used entropy as an initial investigation for the following two reasons.
> - Both the MixUp and the PGD attack used by MixProp and AdvProp are data transformations that increase entropy.
> - Entropy and confidence are the most commonly used baseline indicators in the field of OOD detection research.
>
> We will revise the paper to emphasize these motives.
> Furthermore, the search for metrics more suitable than entropy is a future work.
>
> > provide further results in the distribution shift caused by the proposed method.
>
> We measured FID, a measure of inter-distributional distance between two datasets.
> We used Vanilla-trained ResNet18 on the CIFAR100 dataset.
> We provide FID for the original dataset and the dataset generated by three transformations: MixUp , PGD attack, and Ours (Sample selection + PGD attack + MixUp).
> |     Method                      |       FID     |
> |---------------------------------|:-------------:|
> |     MixUp                       |      5.12     |
> |     PGD                         |     373.93    |
> |     Ours          |     383.89    |
>
> Ours shows the most FID increase and results in the pushing original distribution far away from another.
>
> > If data augmentation results vary, as indicated in Fig. 5, the main results in Tab. 2 and Tab. 4 should include statistical tests.
>
> Thank you for pointing that out.
> We will include the standard errors in Tables 2 and 4.
> Here we show some of the results in Table 2 with standard errors.
>
> | Dataset                  |               | Pets           |                |              | Cars           |                |
> |--------------------------|---------------|----------------|----------------|--------------|----------------|----------------|
> |                          | SA            | RA             | H              | SA           | RA             | H              |
> | Vanilla                  | 92.24±0.08    | 49.97±0.33     | 64.82±0.29     | 90.18±0.14   | 41.07±0.34     | 56.43±0.32     |
> | MixProp                  | **93.01±0.1** | 59.84±0.24     | 72.82±0.16     | **91.3±0.1** | 51.13±0.39     | 65.55±0.35     |
> | Fast AdvProp             | 92.78±0.06    | 54.42±0.85     | 68.6±0.69      | 90.71±0.1    | 44.31±0.39     | 59.54±0.34     |
> | AdvProp                  | 92.1±0.12     | 55.82±0.55     | 69.51±0.46     | 90.45±0.02   | 54.11±0.56     | 67.71±0.44     |
> | EntProp (k = 0.2, n = 1) | 92.23±0.27    | 59.28±0.34     | 72.18±0.24     | 90.48±0.03   | 52.93±0.84     | 66.79±0.66     |
> | EntProp (k = 0.6, n = 1) | 92.21±0.12    | 60.14±0.67     | 72.8±0.52      | 90.36±0.1    | 52±0.39        | 66.01±0.32     |
> | EntProp (k = 0.2, n = 5) | 92.47±0.08    | 62.35±0.12     | 74.48±0.07     | 90.15±0.08   | **55.66±0.67** | **68.82±0.49** |
> | EntProp (k = 0.6, n = 5) | 92.15±0.13    | **66.75±0.22** | **77.42±0.19** | 90.21±0.12   | 52.72±0.11     | 66.55±0.12     |
>
> Even when standard errors are taken into account, EntProp shows superior results.
>
> > Fig. 2 is a bit confusing, since the left and the right both have orange and blue boxes, yet the legends are different, which is not intuitive for comparison. The caption of Fig. 2 has typos. The x-axis in Fig. 6 is confusing.
>
> Thank you for pointing that out.
> We will correct it as you pointed out.

---

### Official Review · Reviewer_dFhD · 2024-03-05

**Q2-1 Originality-Novelty:** 3
**Q2-2 Correctness-Technical Quality:** 3
**Q2-5 Clarity Of Writing:** 3

**Q1 Summary And Contributions:**

This paper proposed EntProp based on an interesting finding that generating the adversarial samples based on the training sample with high entropy could be used to enhance the performance of the adversarial propagation. Following this, the author uses data augmentation and free adversarial technology to improve the efficiency of the adversarial propagation method. Under the same time cost (e.g., 1.2x), the experimental results show the proposed method achieves the state-of-the-art (SOTA) performance.

**Q2-3 Extent To Which Claims Are Supported By Evidence:**

3: Good: the main claims are supported by convincing evidence (in the form of adequate experimental evaluation, proofs, (pseudo-)code, references, assumptions).

**Q2-4 Reproducibility:**

2: Fair: key resources (e.g. proofs, code, data) are unavailable but key details (e.g. proof sketches, experimental setup) are sufficiently well-described for an expert to confidently reproduce the main results.

**Q3 Main Strengths:**

1. The motivation of the paper is solid. The finding is interesting and motivated since the classifier does not well learn samples with high entropy. Therefore, using these samples to generate adversarial samples and to implement the adversarial propagation method is sound.

2. EntProp could achieve SOTA performance while keeping low computation costs.

**Q4 Main Weakness:**

1. The writing for this paper should be improved. For example, the transformed samples used in this paper are confused. Since the transformed samples are all adversarial samples, why not directly use adversarial samples? Meanwhile, it is better to emphasize that this paper is under an adversarial propagation setting, which could eliminate some mistakes.

2. It may be better to report the performance based on the ImageNet-C similar to Fast AdvProp since ImageNet-C is an important OOD dataset.

3. The proposed two techniques seem to the incremental works since these have been used in Fast AdvProp.

**Q5 Detailed Comments To The Authors:**

The main contribution of this paper is the empirical finding instead of the two technologies. It is better to emphasize this part and give more details about entropy. For example it may be better to explore the influence of different adversarial samples generated by other methods such as Autoattack and C&W attack instead of only focusing on PGD.

**Q9 Complying With Reviewing Instructions:**

Yes

---

> ### Author Rebuttal · Authors · 2024-04-07
>
> We would like to thank you for your in-depth understanding of our paper and your thoughtful feedback to improve its quality.
> We appreciate your recognition that our motivation is solid, and EntProp achieve SOTA performance.
>
> > The writing for this paper should be improved. For example, the transformed samples used in this paper are confused. Since the transformed samples are all adversarial samples, why not directly use adversarial samples? Meanwhile, it is better to emphasize that this paper is under an adversarial propagation setting, which could eliminate some mistakes.
>
> Sorry for the confusion.
> We use the term "transformed sample" because our method can be used for transformations that increase entropy as well as adversarial attacks.
> In many experiments our method uses adversarial samples, but in some experiments we do not use adversarial samples (see Sec. 4.2.4).
>
> > It may be better to report the performance based on the ImageNet-C similar to Fast AdvProp since ImageNet-C is an important OOD dataset.
>
> Sorry for the confusion.
> The RA results we show in Table 2 for the ImageNet dataset indicate the accuracy on the ImageNet-C dataset.
> We evaluated RA using the artificial corruptions used in the ImageNet-C dataset against ImageNet and other datasets.
> We will revise our paper so that the presentation is not misleading.
>
> > The proposed two techniques seem to the incremental works since these have been used in Fast AdvProp.
>
> Our main contribution is not technology, but innovative insights and sample selection method based on entropy.
> Our insights contribute to the understanding of the disentangled learning via ABNs, which is not fully understood why it works.
> Our insight-based method, EntProp, is technically simple but achieves sufficient performance gains.
> We argue that simple technology is rather beneficial because it is easier to maintain and implement and more practical.
>
> > The main contribution of this paper is the empirical finding instead of the two technologies. It is better to emphasize this part and give more details about entropy.
>
> We investigated the effect of increased entropy on the dataset distribution by measuring FID.
> Please see rebuttal to reviewer 5ua7 for results.
>
> > For example it may be better to explore the influence of different adversarial samples generated by other methods such as Autoattack and C&W attack instead of only focusing on PGD.
>
> We investigated the influence of adversarial attacks other than PGD.
> The AdvProp paper experimented with the PGD variants GD and I-FGSM and concluded that the type of attack had no effect on performance.
> We experimented with C&W and TRADES, which are different types of attacks from PGD.
> TRADES is a typical attack for adversarial training and uses cross-entropy loss and KL divergence loss.
> We could not experiment with Autoattack because of the huge amount of computation time involved.
> We trained ResNet50 on the CIFAR100 dataset with EntProp (k=0.6, n=5) using different attacks.
> We show the results in the table below.
>
> | Adversarial Attack |     SA    |     RA    |     H     |
> |--------------------|:---------:|:---------:|:---------:|
> | Ours (PGD)         | **80.62** | **60.50** | **69.12** |
> | C&W                |   79.93   |   55.86   |   65.76   |
> | TRADES             |   80.07   |   58.25   |   67.44   |
>
> C&W and TRADES results are inferior to PGD for both SA and RA.
> CW is an attack designed to increase DNN misclassification and TRADES is an attack designed to balance DNN accuracy and adversarial robustness, and has a smaller effect on increasing entropy than PGD.
> Thus, this result confirms the validity of our method design, which uses transformations that increase entropy.

---

### Official Review · Reviewer_rCNi · 2024-03-09

**Q2-1 Originality-Novelty:** 2
**Q2-2 Correctness-Technical Quality:** 3
**Q2-5 Clarity Of Writing:** 3

**Q1 Summary And Contributions:**

The authors propose a method that incrementally builds on existing OOD generalization techniques such as AdvProp and Fast AdvProp that use disentangled batch norm stats / layers for in-distribution and OOD domains to leverage adversarial samples to improve accuracy and robustness. The contributions include a novel adaptation of existing methods that uses entropy as a way to disambiguate between samples in and out of distributions using which they are able to obtain higher performance and better generalization than baseline methods, and an empirical validation of their approach.

**Q2-3 Extent To Which Claims Are Supported By Evidence:**

3: Good: the main claims are supported by convincing evidence (in the form of adequate experimental evaluation, proofs, (pseudo-)code, references, assumptions).

**Q2-4 Reproducibility:**

2: Fair: key resources (e.g. proofs, code, data) are unavailable but key details (e.g. proof sketches, experimental setup) are sufficiently well-described for an expert to confidently reproduce the main results.

**Q3 Main Strengths:**

1. The paper is well written and easy to follow.
2. The related works are very well researched and built upon. Strengths of existing works seem to be appropriately incorporated.
3. The paper includes an analysis of the training cost of the method which is helpful for practical applications in the real world. The results section is quite comprehensive - the paper includes comparisons against several baselines, and ablation experiments that increase trust in the effectiveness of the proposed method.
4. The paper also calls out known limitations of the proposed method in that suitable metrics to differentiate between domains must be used for different objectives.

**Q4 Main Weakness:**

1. The experiments are quite comprehensive and can further be strengthened by including comparisons against different adversarial attacks or augmentation strategies, or even different combinations of these. How does the proposed method compare or work with AutoAugment or RandAugment to find augmentation policies from the data. How well does free adversarial training generalize across different kinds of adversarial attacks or perturbations?
2. How would the method disambiguate between more than 2 domains, for example if there are multiple training data sources such as clean, augmented and adversarial examples? Would it help to differentiate between >2 domains using metrics such as entropy and how does the method compare against AdvProp which can maintain multiple ABNs?
3.The perturbations supported by the framework are limited and tailored for computer vision tasks. It would be useful to discuss the generalization of the solution to other input modalities and perturbations.

**Q5 Detailed Comments To The Authors:**

Overall, the paper reads quite well - but I would like to see a better motivation for the proposed method, e.g. how do clean and transformed samples change wrt the domain distribution for metrics other than entropy?
Questions:
-How would the method disambiguate between more than 2 domains, for example if there are multiple training data sources such as clean, augmented and adversarial examples? Would it help to differentiate between >2 domains using metrics such as entropy and how does the method compare against AdvProp which can maintain multiple ABNs?
-The perturbations supported by the framework are limited and tailored for computer vision tasks. It would be useful to discuss the generalization of the solution to other input modalities and perturbations.
-Is there a qualitative understanding of the metric that should be considered for domain disambiguation? Or corresponding augmentations or adversarial attacks to use to push samples further out of domain for those metrics.

Minor comments
-OOD robustness may be a rather general notion, it may help to clarify what's in scope for "in-domain" or out of domain samples. A more concrete definition of distribution shifts may also be helpful.

**Q9 Complying With Reviewing Instructions:**

Yes

---

> ### Author Rebuttal · Authors · 2024-04-08
>
> We would like to thank you for your in-depth understanding of our paper and your thoughtful feedback to improve its quality.
> We appreciate your recognition that our paper reads quite well.
>
> > ... different adversarial attacks or augmentation strategies, or even different combinations of these. ... compare or work with AutoAugment or RandAugment to find augmentation policies from the data.
>
> Comparison with AutoAugment is discussed in the AdvProp paper.
> The larger the model size (EfficientNetB6 or larger), the higher the gain of AdvProp over AutoAugment, and the accuracy is further improved when these techniques are used together.
> We compared RandAugment (N=2, M=7, remove duplicate augmentations to corrupted dataset) and EntProp.
> We trained ResNet18 on the CIFAR100 dataset.
> We show the results in the table below.
>
> |                                    |   SA  |   RA  |   H   |
> |------------------------------------|:-----:|:-----:|:-----:|
> | Vanilla                            | 78.45 | 49.96 | 61.04 |
> | EntProp(k=0.6, N=5)                | **78.92** | 57.16 | **66.30** |
> | Vanilla w/RandAugment               | 77.29 | 53.74 | 63.40 |
> | EntProp(k=0.6, N=1) w/RandAugment   | 77.02 | **57.54** | 65.87 |
> | EntProp(k=0.6, N=5) w/RandAugment   | 76.64 | 56.99 | 65.37 |
>
> The comparison results with RandAugment show that EntProp outperforms both SA and RA.
> The combination of RandAugment and EntProp is worse than normal EntProp.
> This result may be due to a lack of tuning of RandAugment's hyperparameters or a possible conflict with MixUp.
> In addition, as the AdvProp paper discusses, increasing the model size may solve the problem.
>
> > ... free adversarial training generalize across different kinds of adversarial attacks or perturbations?
>
> We experimented with TRADES, a different kind of adversarial attack than PGD, by introducing free adversarial training.
> We trained ResNet50 on the CIFAR100 dataset.
> We show the results in the table below.
>
> | Adversarial Attack | Cost |     SA    |     RA    |     H     |
> |--------------------|------|:---------:|:---------:|:---------:|
> | TRADES             | 7N   | **80.07** | **58.25** | **67.44** |
> | TRADES w/Free adversarial training | 6N   |   79.81   |   58.13   |   67.27   |
>
> Free adversarial training reduces accuracy slightly, but also reduces training cost.
> For TRADES, which minimizes Cross-entropy and KL-divergence, free adversarial training using only Cross-entropy is not optimal, but it is effective enough in our research context.
> Thus, free adversarial training generalizes across different kinds of adversarial attacks.
>
> > ... disambiguate between more than 2 domains ...
>
> We distinguish clean, augmented and adversarial samples into two domains In-distribution/Out-of-distribution based on entropy.
> In the study of adversarial robustness, a method is proposed to train the four domains of cln, ℓ1, ℓ2, and ℓ∞ perturbations via a gated multi-BN layer (https://arxiv.org/abs/2012.01654).
> It may be promising to introduce this method into our research context and distinguish more than three domains.
> However, the computational cost of gated multi-BN layers is very high, so solving this problem is a future work.
>
> > ...  the generalization of the solution to other input modalities and perturbations.
>
> As shown in the rebuttal to e5zd, EntProp is also applicable to ViT, a common architecture for various modalities.
> EntProp is applicable to any modality where uncertainty metrics and data transformations that increase uncertainty, such as adversarial attacks, are existing.
>
> > ...  better motivation for the proposed method, e.g. how do clean and transformed samples change wrt the domain distribution for metrics other than entropy?
>
> We also performed the experiment shown in Figure 1 for confidence/cross entropy/logit mergin.
> Since these metrics have almost the same meaning, the results also show almost the same trend.
> We cannot include the figure here, so we include it in the appendix.
>
> With respect to the motivation of the proposed method, we added an experiment of FID measurements.
> Please see the rebuttal to 5ua7 for the results.
>
> > Is there a qualitative understanding of the metric ... Or corresponding augmentations or adversarial attacks to use to push samples further out of domain for those metrics.
> > Minor comments ... clarify what's in scope for "in-domain" or out of domain samples. A more concrete definition of distribution shifts may also be helpful.
>
> In general, " in distribution" ("in domain") refers to the distribution in which the DNN has been trained.
> Therefore, we think that metrics such as entropy calculated from a DNN that is being trained or has been trained is suitable.
> Similarly, adversarial attacks such as PGD that use the uncertainty of the DNN during training are appropriate in our research context.
> In fact, PDG had better accuracy results than other adversarial attacks whose goal is other than uncertainty increase (please see the rebuttal to dFhD).

---

### Official Review · Reviewer_e5zd · 2024-03-13

**Q2-1 Originality-Novelty:** 2
**Q2-2 Correctness-Technical Quality:** 3
**Q2-5 Clarity Of Writing:** 3

**Q10 Ethical Concerns:**

No potential ethical concern.

**Q1 Summary And Contributions:**

This paper proposes the Entprop algorithm to improve classification accuracy and robustness on out-of-distribution data.
Specifically, from the perspective of entropy for uncertainty measurement, transformed data can be in-domain or out-of-domain when compared with clean data. Based on this observation, with a reasonable threshold, clean samples with high entropy are selected. Then, adversarial examples of these selected samples are treated as out-of-distribution data.
During training, these out-of-distribution data use additional BatchNorm layers.

Experiments on ImageNet and CIFAR show some improvements over baselines, like MixProp, and Fast AdvProp.

**Q2-3 Extent To Which Claims Are Supported By Evidence:**

2: Fair: the main claims are somewhat supported by evidence (but the experimental evaluation may be weak, or does not match entirely with the claims, important baselines may be missing, proofs contain important ideas but lack rigor, algorithmic details are only discussed superficially, references are imprecise, assumptions are not sufficiently motivated or explicated, etc.).

**Q2-4 Reproducibility:**

2: Fair: key resources (e.g. proofs, code, data) are unavailable but key details (e.g. proof sketches, experimental setup) are sufficiently well-described for an expert to confidently reproduce the main results.

**Q3 Main Strengths:**

Strengths:
(1) The paper is clear and easy to follow.
(2) The proposed algorithm is simple to implement and seems to be effective.

**Q4 Main Weakness:**

(1) The proposed algorithm is based on the BN-based networks. Would it be applicable to recent vision transformers?
(2) Recent work, like MAE [1], and GPaCo[2] can improve both accuracy and robustness. Is the proposed method complementary to them?

[1] Masked Autoencoders Are Scalable Vision Learners. CVPR 2022.
[2] Generalized Parametric Contrastive Learning. TPAMI 2023.

(3) ImageNet-C is the most popular benchmark for robustness on out-of-distribution data.
For Table 3, the results of standard accuracy and robustness under ImageNet-C are missing.

(4) The results in Table 5 seem to be wired. Under the setting without mixup, compared with Fast AdvProp results (SA 78.89\%, RA 53.31\%) in Table 4, the proposed method only achieves inferior accuracy and robustness (SA 78.55\%, RA 52.99\%) shown in Table 5. Do the claimed improvements only come from the mixup?

(5) Sound ablations should be included to validate the effectiveness of the proposed Entprop algorithm. For example, will Fast AdvProp/advProp be improved when combined with Entprop?

**Q5 Detailed Comments To The Authors:**

(1) More ablations should be designed to validate the effectiveness of the proposed algorithm.
(2) The authors should show the flexibility of the proposed method and validate that it is complementary to previous methods.

**Q9 Complying With Reviewing Instructions:**

Yes

---

> ### Author Rebuttal · Authors · 2024-04-08
>
> We would like to thank you for your in-depth understanding of our paper and your thoughtful feedback to improve its quality.
> We appreciate your recognition that our paper is clear, and EntProp is simple yet effective.
>
> > (1) ... Would it be applicable to recent vision transformers?
>
> > (2) Recent work, like MAE [1], and GPaCo[2] ... complementary to them?
>
> Thank you for introducing the interesting papers.
> We experimented with fine-tuning ViT-Base pre-trained by MAE on the CIFAR100 dataset.
> We used CE or GPaCo as the loss function for fine-tuning.
> When applying EntProp to ViT we add an auxiliary LN layer instead of an auxiliary BN layer.
> We show the results in the table below.
>
> |  Loss | EntProp |     SA    |     RA    |     H     |
> |:-----:|:-------:|:---------:|:---------:|:---------:|
> |   CE  |         |   89.36   |   70.62   |   78.89   |
> |   CE  |    x    |   89.37   | **72.57** | **80.10** |
> | GPaCo |         | **89.73** |   70.23   |   78.79   |
> | GPaCo |    x    |   89.37   |   70.92   |   79.08   |
>
> The results show that H improves when used in conjunction with EntProp, regardless of whether the loss function is CE or GPaCo.
> In other words, EntProp can be applied to ViT-based architectures and can be used in conjunction with recent methods to improve SA and RA.
>
> > (3) ... ImageNet-C are missing.
>
> Sorry for the confusion.
> The RA results we show in Table 2 for the ImageNet dataset indicate the accuracy on the ImageNet-C dataset.
> We evaluated RA using the artificial corruptions used in the ImageNet-C dataset against ImageNet and other datasets.
> We will revise our paper so that the presentation is not misleading.
>
> > (4) ... Do the claimed improvements only come from the mixup?
>
> As pointed out, the gain from MixUp is significant, but EntProp's improvement does not only come from MixUp.
> We used Fast AdvProp and MixUp to train the model on the CIFAR100 dataset and compare it to EntProp.
> We show the results in the table below.
>
> |                     |    R18    |     　    |     　    |    WRN    |    　    |   　   |    RXT    |     　    |     　    |
> |---------------------|:---------:|:---------:|:---------:|:---------:|:--------:|:------:|:---------:|:---------:|:---------:|
> | Method              |     SA    |     RA    |     H     |     SA    |    RA    |    H   |     SA    |     RA    |     H     |
> | Fast AdvProp (w/MixUp)  |   78.94   |   54.65   |   64.59   |   80.19   |   57.16  |  66.74 |   80.74   |   57.83   |   67.39   |
> | EntProp(k=0.2, n=1) | **79.41** | **55.24** | **65.15** | **80.66** | **57.3** | **67** | **81.46** | **58.47** | **68.08** |
>
> In all models, EntProp outperforms Fast AdvProp in SA and RA.
> The most significant difference between EntProp and Fast AdvProp is sample selection.
> EntProp is entropy-based, while Fast AdvProp randomly selects samples.
> Entropy-based sample selection works well together because it further accelerates the entropy increase due to MixUp.
> Thus, the improvement we claimed comes from MixUp and entropy-based sample selection.
>
> > (5) ... will Fast AdvProp/advProp be improved when combined with Entprop?
>
> EntProp can be combined with Fast AdvProp/Advprop to improve accuracy.
> We experimented with the combination of Fast AdvProp and EntProp (k=0.2) with ResNet18 on the CIFAR100 dataset.
> When combined with EntProp, Fast AdvProp uses entropy-based sample selection instead of random sample selection.
> We show the results in the table below.
>
> | Method        | Cost |     SA    |     RA    |     H     |
> |---------------|:----:|:---------:|:---------:|:---------:|
> | Vanilla       |   N  |   78.45   |   49.96   |   61.04   |
> | Fast AdvProp | 1.2N |   78.89   |   53.31   |   63.63   |
> | Fast AdvProp w/EntProp | 1.4N | **79.39** | **55.20** | **65.12** |
>
> Combination with EntProp slightly increases computational cost due to entropy calculation overhead, but also increases accuracy.
> However, it is inferior to EntProp (1.2N) (see table above), thus our method design is superior.
>
> Next, we experimented with the combination of AdvProp and EntProp (k=0.6) with ResNet50 on the CIFAR100 dataset.
> When combined with EntProp, AdvProp uses pure PGD attack without Free Adversarial Training.
> We show the results in the table below.
>
> | Method             |      |     SA    |     RA    |     H     |
> |--------------------|:----:|:---------:|:---------:|:---------:|
> | Vanilla            |   N  |    79.3   |   51.01   |   62.08   |
> | AdvProp            |  7N  |   78.05   |   58.94   |   67.17   |
> | AdvProp w/EntProp | 4.6N | **80.85** |   **60.68**   | **69.33** |
>
> EntProp reduces the cost of AdvProp and improves accuracy.

---

### Meta-Review · Area_Chair_sUK1 · 2024-04-17

The overall consensus seems to be that this work is well-presented and practical approach to improving robustness across domains, and I agree.  The reviewers have given a number of suggestions of how to further strengthen the experimental results, and I believe that the authors have done a convincing job of alleviating reviewer concerns in the rebuttal.  I encourage the authors to incorporate these suggestions into the draft.